# Handgrip Strength and Its Relationship with White Blood Cell Count in U.S. Adolescents

**DOI:** 10.3390/biology10090884

**Published:** 2021-09-08

**Authors:** José Francisco López-Gil, Robinson Ramírez-Vélez, Mikel Izquierdo, Antonio García-Hermoso

**Affiliations:** 1Departamento de Expresión Plástica, Musical y Dinámica, Facultad de Educación, Universidad de Murcia (UM), 30100 Murcia, Spain; josefranciscolopezgil@gmail.es; 2Health and Social Research Center, Universidad de Castilla-La Mancha (UCLM), 16071 Cuenca, Spain; 3Navarrabiomed, Complejo Hospitalario de Navarra (CHN), Universidad Pública de Navarra (UPNA), IdiSNA, 31008 Navarra, Spain; robin640@hotmail.com (R.R.-V.); mikel.izquierdo@gmail.com (M.I.); 4El Deporte y la Salud, Escuela de Ciencias de la Actividad Física, Universidad de Santiago de Chile (USACH), Santiago 71783-5, Chile

**Keywords:** muscular fitness, physical fitness, immune system, youths

## Abstract

**Simple Summary:**

Despite the availability of scientific evidence demonstrating the relationship between higher physical fitness levels and a healthy status (e.g., lower adiposity and cardiovascular risk, greater bone health) in children and adolescents, the association between the total count of white blood cell (WBC) and groups based on handgrip strength and body composition remains unclear. We found a positive association between low handgrip strength (for all groups established) and higher WBC count (in both sexes). Furthermore, those with low handgrip strength (in all estimations) showed greater WBC count in both boys and girls. This finding contributes to the scientific literature supporting the significance of promoting muscular fitness in adolescents.

**Abstract:**

Background: The role of muscular fitness (as a protecting factor for an optimal immune system) and WBC count remains unclear. To the best of our knowledge, this is the first study to investigate the relationship between the total count of WBC and groups based on handgrip strength and body composition. The aim of this study was to elucidate the relationship between handgrip strength and WBC count in a nationally representative sample of adolescents from the U.S. Methods: We used data from the NHANES cross-sectional study (waves 2011 to 2014). The final analysis included 917 adolescents from 12–17 years old (51.0% boys). Normalized handgrip strength (kg) (NHS) was relativized by body composition parameters (body weight [NHSw], total body fat [NHSf], and trunk fat [NHSt]) assessed with dual energy X-ray absorptiometry. Results: An inverse association was found between total WBC count and all assessments of low NHS in both sexes (*p* < 0.05). Both boys and girls with low NHS had higher WBC counts in all estimations of NHS than those with high NHS (*p* < 0.05 for all). All estimations of low NHS showed significant differences with those with intermediate NHS (only in girls) (*p* < 0.05 for all). Girls with intermediate NHSt exhibited higher WBC count than those with high NHSt (*p* = 0.004). Conclusions: Our findings suggest that greater levels of NHS are linked to lower total WBC counts. From a public health perspective, our findings are clinically significant and emphasize the relevance of improving muscular fitness during adolescence since it may contribute to boosting the immune system among adolescents.

## 1. Introduction

The white blood cell (WBC) count shows the number of leukocytes (i.e., neutrophils, basophils, eosinophils, monocytes and lymphocytes) in a blood test. The total WBC count is a helpful clinical marker and is frequently used as a diagnostic tool for adults, children and adolescents [1]. High or low total WBC counts flowing in a state of rest may be related to the existence of illness, such as infection or inflammatory processes [2]. Likewise, normal-range count of WBC’s, which are produced in the bone marrow (granulocytes) and in the lymphoid organs (lymphocytes), exert protection over the human body against infections and aid in the immune response [3].

Scientific evidence has extensively exhibited the associations between higher physical fitness levels and a healthy status in children and adolescents [4,5]. For instance, muscular fitness has been linked to lower adiposity and cardiometabolic risk parameters later in life, as well as has with a positive association with bone health (referring to mineral component) in this population [4]. Additionally, increasing evidence emphasizes the function of muscular fitness as a protective factor for cardiometabolic parameters during adulthood [6].

According to the relationship between the immune system and muscular fitness, a recent systematic review and meta-analysis performed by Tuttle et al. [7] indicated that higher systemic inflammation is linked to lower muscle strength and muscle mass in humans. Conversely, only a few studies [8,9,10] have examined the relationships between WBC counts and cardiorespiratory fitness, physical activity and sedentary behavior among children and adolescents. Nevertheless, the role of muscular fitness (as a protecting factor for an optimal immune system) and WBC remains unclear. In this sense, to the best of our knowledge, this is the first study to investigate the relationship between the total count of WBC and groups based on handgrip strength and body composition. Therefore, the aim of this study was to elucidate the relationship between handgrip strength and WBC counts (i.e., basophils, eosinophil neutrophils, monocytes and lymphocytes) in a nationally representative sample of adolescents from the U.S. Our hypothesis is that higher relative handgrip strength could be linked to lower WBC counts.

## 2. Materials and Methods

### 2.1. Study Design

NHANES data from the 2011 to 2014 wave were used in this study. A simple stratified multi-stage probability sampling of non-institutionalized civilian residents of the U.S. was performed to collect the data. Based on the initial 19,931 data records, 3709 participants were withdrawn because they were younger than 6 years old (missing handgrip strength data); 11,977 were dropped because they were younger than 12 (no physical activity data) or older than 18 years (adults); 2355 adolescents were excluded due to missing values of parameters of interest (body fat composition, handgrip strength, WBC, etc.); and lastly, 1085 were removed due to lack of data on some covariates. Adolescents with asthma were removed from the analyses because WBC count is influenced by this disease (*n* = 275). Moreover, since the relationship between handgrip strength and WBC count is not linear, adolescents with a WBC count below 4000 cells per microliter of blood (leukopenia) (*n* = 57) and above 11,000 cells per microliter of blood (leukocytosis) (*n* = 39) were excluded. Finally, data from 917 remaining adolescents (aged 12–17) were included in the final analyses.

The ethical standards of the institutional and/or national research committee [National Health and Nutrition Examination Survey, NCHS IRB/ERB Protocol Number: NHANES 2011–2012 (Protocol #2011-17); NHANES 2013–2014 (Continuation of Protocol #2011-17)] were followed for all the procedures with human beings. As the data did not include personal identifiers, no additional consent was required for this study.

### 2.2. Procedures

#### 2.2.1. Anthropometric Data and Body Composityion

Body weight was obtained by a digital weight scale. A stadiometer was used to determine standing height. The formula weight (kg)/height (m^2^) was applied to obtain the body mass index (BMI). Likewise, the Centers for Disease Control and Prevention criteria for assessing excess weight (overweight/obesity) were considered [11].

Dual energy X-ray absorptiometry (DXA) (Hologic QDR 4500A) was used to compute whole body fat and trunk fat. This technology uses two different energy levels created by an energy tube to assess bone mineral content and bone mineral density. A certified radiology technologist performed all DXA scans.

#### 2.2.2. Blood Extraction

Participants were evaluated in the morning after fasting for at least 9 h. For the blood extraction (venipuncture) procedure, some conditions were required, such as completing a questionnaire to identify conditions that would warrant excusing the participant from blood collection, checking the fasting status, and executing the blood extraction.

The amount of each WBC subtype (i.e., neutrophils, eosinophils, basophils, monocytes and lymphocytes; cells/μL) was calculated through the Beckman Coulter® count and sizing technique, in conjunction with an automatic diluting and mixing device for the processing of the samples and a single-beam photometer for hemoglobinometry. The phlebotomist extracted a 3- or 5-mL into a K3 EDTA tube from all the youths on the basis of the recognized venipuncture protocol and practices (a 1–2% dilution effect appears in this liquid EDTA tube).

#### 2.2.3. Handgrip Strength

A handgrip dynamometer (T.K.K. 5401, Grip-D, Takei, Japan) adjusted by hand size for each participant was used to evaluate handgrip strength. Adolescents stood with their arms outstretched and gradually and continuously squeezed the dynamometer to their maximum strength (at least 2 s). The test was carried out twice interchanging with both hands with a rest frame of 90 s between the different tries [12]. The mean of the left and right hands was computed as absolute handgrip strength (kg) and then normalized (NHS) to body composition parameters (by body weight [NHSw], by total body fat [NHSf], and by trunk fat [NHSt]). Due to the lack of cut-off points for NHSw, NHSf and NHSt, the different estimations of these NHS values were divided into terciles by age and sex for all the samples and classified as high handgrip strength (first tercile), intermediate handgrip strength (second tercile), and low handgrip strength (third tercile). A further description of the testing procedures can be found in the NHANES Muscle Strength Procedures Manual [13].

#### 2.2.4. Covariates

Age, sex, the ratio of family income to poverty, and race/ethnicity were self-reported in adolescents over 16 years old, as well as in adolescents who were emancipated. Participants were asked to answer questions designed to assess the time spent in physical activity, both moderate physical activity and vigorous physical activity in three different scenarios (work, recreation, and transportation). Therefore, the overall weekly metabolic equivalent for task (MET)-minutes were determined based on the guidelines of the NHANES National Youth Fitness Survey Time (vigorous physical activity = 8 METs; moderate physical activity and transportation = 4 METs). Similarly, participants were also asked to answer questions related to the time spent in sitting (at school, at home), getting to and from places, or with friends including time spent sitting at a desk, traveling in a car/bus, playing cards, reading, using computers, or watching TV (in minutes). Dietary intake was calculated by two 24-h dietary recall in-person interviews in person for each participant. A further extensive description of the quality and control methods can be found on the NHANES website (http://www.cdc.gov/nchs/nhanes.htm; accessed on 6 January 2021).

### 2.3. Statistical Analysis

Descriptive information is depicted as numbers and percentages (*n* [%]) for categorical variables and the mean and standard deviation for continuous variables. Chi-square tests and Student’s *t* tests were carried out to compare categorical and continuous variables, respectively. All analyses were performed stratified by sex since preliminary analyses showed differences in WBC count in the interaction between sex and handgrip strength groups (all *p* values < 0.05). Similarly, this choice was followed the recommendation to take sex into account when performing WBC analyses [14]. To establish the association between handgrip strength variables and WBC count, analysis of covariance (ANCOVA) was applied. As the WBC count did not show a normal distribution, the assumptions for performing an ANCOVA were not met. Therefore, we selected bootstrapping as a reliable technique to establish robust assessments of standard errors, as well as confidence intervals for measures of both central tendency and association. Therefore, robust bootstrapping ANCOVA was carried out with significance set at *p* < 0.05 and with adjustment for confounding variables. This analysis was also conducted to evaluate the differences between the mean values of WBC count across the established handgrip strength groups (high, intermediate, or low) normalized by different body composition parameters, as previously mentioned. Additionally, to confirm the differences between the mean values of WBC count across handgrip strength groups, Bonferroni pairwise post hoc comparisons were conducted. All analyses performed were adjusted by age, the ratio of family income to poverty, race/ethnicity, physical activity and dietary intake. Likewise, we applied the survey commands in STATA 16.1 (StataCorp, College Station, TX, USA) to conduct all analyses to consider the weighting for each observation. The statistical significance was maintained at *p* < 0.05.

## 3. Results

Descriptive information of the analyzed sample is depicted in Table 1. The final sample had a mean age (standard deviation [SD]; [range]) of 14.5 years (1.7) with a range from 12 to 17 years old (51.0% boys). Girls had greater levels of neutrophils and higher WBC counts (*p* < 0.001). Conversely, boys had higher levels of eosinophils than girls (*p* < 0.001). According to handgrip strength, differences were found in all assessments of handgrip strength between the sexes (absolute handgrip strength, NHSw, NHSf and NHSt) (boys were stronger than girls, *p* < 0.001). Last, the prevalence of overweight/obesity was 39.1% for the whole sample, based on the CDC criteria.

The association between different assessments of NHS and total WBC count, segmented by sex is depicted in Table 2. A positive association was found between total WBC count and all assessments of low NHS in both boys and girls (*p* < 0.05). Similarly, a positive association was shown between the total WBC concentration and intermediate NHSf and NHSt (only in the case of girls) (*p* < 0.05).

Figure 1 presents the differences in the mean values of total WBC count according to the handgrip strength groups by sex. Both boys and girls with low NHS had higher values of total WBC count in all estimations of NHS than those with high NHS (*p* < 0.05). Additionally, statistical significance was also found between all the estimations of low NHS and intermediate NHS (*p* < 0.05) (only in girls). Additionally, in NHSt, significant differences were found between intermediate NHSt and high NSHt (*p* = 0.004).

## 4. Discussion

The main aim of this study was to clarify the association between NHS and total WBC count in a nationally representative sample of U.S. adolescents, which showed a negative relationship. This association was corroborated for the three different handgrip strength indicators applied (NHSw, NHSf, and NHSt). Our results strengthen the current public health physical activity guidelines (which highlight the importance of engaging in muscle strengthening activities for the first time) [15], suggesting that higher levels of NHS could be related to the immune system of adolescents. This fact might be particularly significant among those with low-grade systemic inflammation [16].

There are different physiological pathways by which increased handgrip strength could influence the WBC concentration. First, a recent systematic review and meta-analysis found that greater handgrip strength (and knee extension strength) is related to lower pro-inflammatory cytokine levels, such as interleukin-6 (IL-6) and tumor necrosis factor-α (TNF-α) [7], which boost neutrophils, possibly via cortisol and the hypothalamic-pituitary axis [17]. Notwithstanding, it has also been indicated that glucocorticoids may exert a protective influence on neutrophils survival by delaying apoptosis [18]. Second, handgrip strength appears to be inversely associated with leptin concentrations in youth, both independently and globally [19], which stimulates leukocytes [20] and promotes the release of pro-inflammatory cytokines, such as TNF-α and IL-6 [21]. Third, regular physical activity and higher handgrip strength levels could promote variations in neutrophil transfer between bone marrow and the bloodstream, as well as fluctuations in hematopoiesis in bone marrow [22]. Fourth, handgrip strength is related to insulin resistance and glucose metabolism in adolescents [23] and decreased reactive oxygen species [24], all well-known key regulatory elements in inflammation processes, skeletal muscle activity and metabolism. However, the possible mediators of this influence have yet to be established [16]. In this line, we applied NHS (adjusted by body mass, body fat mass, and trunk fat mass) in our analyses, and those with lower adiposity may have re-established the balance in the turnover of adipokines such as leptin, which supports the restoration of the regular blood circulation of neutrophils [25]. Therefore, the influence of higher handgrip strength on body fat, IL-6, leptin and TNF-α levels could be responsible for the decrease in the total WBC count described in the current study.

On the other hand, another plausible explanation for these results could lie in the association between muscular fitness and leukocyte telomere length. In this sense, a higher muscular strength (upper body) was related to longer leukocyte telomere length, independent of several possible confounders (e.g., age and physical activity) [26]. Similarly, it has been suggested that longer midlife leukocyte telomere length is linked to increased handgrip strength in the future [27]. Supporting our hypothesis, Mazidi et al. [28] showed a significant negative association between leukocyte telomere length and blood count parameters (e.g., monocyte count) in a sample of U.S. adults. In addition, overweight and obesity have been related to shorter leukocyte telomere lengths, and at the same time, these excess weight statuses have been related to shorter leukocyte telomere lengths in children and adolescents. For this reason, these findings could be relevant since increased BMI in childhood and adolescence might be linked to faster ageing and could have a harmful impact on future health later in life [29]. Conversely, another study using the present sample showed an inverse relationship between WBC count, cardiorespiratory fitness and habitual physical activity levels [10]. People who were physically active or had high aerobic fitness potentially participated in activities that benefited their muscular fitness levels [30], which could explain (at least partially) our results. For this reason, we adjusted the analyses by physical activity level to control for the possible confounding effect of this variable. Thus, our results emphasize not only that it is important to maintain adequate levels of physical activity but also that young people should engage in muscle strengthening activities, as previously mentioned [15].

The current study has some strengths, including that, to our knowledge, this is the first study to explore the association between total WBC count according to handgrip strength and body composition. Furthermore, it is worth mentioning the large and representative sample of adolescents from the U.S. that was analyzed. Likewise, different tools were applied to calculate body composition (e.g., DXA, considered the gold standard method). Conversely, some limitations in this study must be declared, and caution should be taken in the interpretation of these results. First, as our study has a cross-sectional design, it is not possible to establish causality. Second, it is likely that further pro-inflammatory cytokines (e.g., TNF-α, IL-6 or IL-1beta) will be important to incorporate in future studies with the aim of obtaining a full depiction of our results. Third, other categories of lymphocytes (e.g., natural killer cells, B cells, and T cells) and monocytes (i.e., intermediate, non-classical and classical) were not detected, all of which have specific implications in the immune system. Fourth, we did not have access to C-reactive protein values, which could influence the findings [31]. Lastly, the count of WBC is usually a marker of acute, not chronic, inflammation. In this sense, the levels of leukocytes are highly variable (e.g., the half-life of a neutrophil is around 6-8 h in blood [32]). Given these limitations, it is necessary to be cautious in interpreting our results.

## 5. Conclusions

In conclusion, our findings suggest that greater levels of relative handgrip strength are related to lower total WBC counts. Our study emphasizes the relevance of improving muscular fitness during adolescence since it may contribute to boosting the immune system among youths. Furthermore, the results are of interest to physical educators, fitness professionals, trainers, researchers, physical therapists and coaches for detecting the population of concern for primary prevention and for determining the prevalence of adolescents with low or high muscular fitness [33].

## Figures and Tables

**Figure 1 biology-10-00884-f001:**
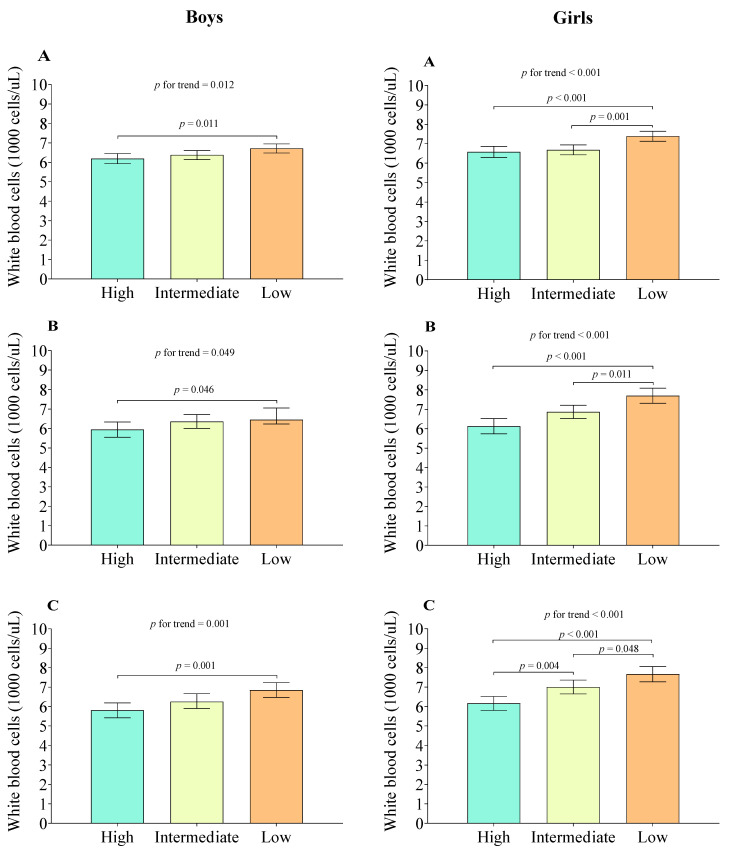
Differences in mean values of total white blood cells count according to the different types of relative handgrip strength and adjusted for age, race/ethnicity, ratio of family income to poverty, physical activity, sedentary behavior and dietary intake. Data expressed as mean (bars) and standard error (lines). (**A**): Normalized handgrip strength by body weight; (**B**): Normalized handgrip strength normalized by body whole body fat; (**C**): Normalized handgrip strength normalized by trunk fat.

**Table 1 biology-10-00884-t001:** Participants’ characteristics of the analyzed sample (*n* = 917).

Variables	Boys		Girls		
	*n*	M (SD)/*n* (%)	*n*	M (SD)/*n* (%)	*p*
Sociodemographic					
Age, years	469	14.5 (1.7)	448	14.6 (1.7)	0.414
Race, Non-Hispanic White, *n* (%)	469	124 (26.4)	448	104 (23.2)	0.068
Family income to poverty ratio	469	2.11 (1.59)	448	2.18 (1.54)	0.455
Dietary intake					
Energy intake (kcal)	469	2199.0 (795.1)	448	1709.2 (623.0)	<0.001
Carbohydrates (g)	469	283.8 (105.4)	448	225.8 (83.9)	<0.001
Proteins (g)	469	84.1 (39.0)	448	62.3 (24.6)	<0.001
Fats (g)	469	82.7 (36.5)	448	63.9 (29.3)	<0.001
Physical Activity					
Weekly PA (MET-min)	469	3263.1 (3012.2)	448	2943.2 (2975.9)	0.092
Daily PA energy expenditure (kcal)	469	541.7 (523.1)	448	456.0 (531.3)	0.011
Sedentary behavior					
Daily sedentary activities (min)	469	504.4 (158.1)	448	540.5 (160.5)	0.001
Anthropometric data					
Weight, kg	469	67.15 (19.75)	448	61.65 (17.71)	<0.001
Height, cm	469	168.5 (9.7)	448	159.7 (6.53)	<0.001
BMI, kg/m^2^	469	23.40 (5.70)	448	24.02 (6.11)	0.095
Overweight/Obese, *n* (%)	469	190 (40.5)	448	172 (38.4)	0.156
Body Fat, kg	199	16.65 (9.36)	205	21.12 (10.04)	<0.001
Body Fat, %	199	24.13 (7.53)	205	33.80 (6.38)	<0.001
Trunk fat, kg	210	6.63 (4.81)	205	8.53 (5.01)	<0.001
Trunk fat, %	210	9.17 (3.96)	205	13.34 (3.94)	<0.001
Muscular fitness					
Absolute Handgrip Strength, kg	469	32.80 (9.10)	448	24.83 (4.91)	<0.001
Handgrip Strength, kg/Body weight, kg	469	0.51 (0.12)	448	0.42 (0.09)	<0.001
Handgrip Strength, kg/Whole body fat, kg	199	2.37 (1.06)	205	1.33 (0.51)	<0.001
Handgrip Strength, kg/Trunk fat, kg	210	6.68 (3.27)	205	3.60 (1.69)	<0.001
Blood test					
Lymphocytes (1000 cells/μL)	469	2.21 (0.62)	448	2.22 (0.62)	0.870
Monocytes (1000 cells/μL)	469	0.53 (0.19)	448	0.52 (0.18)	0.439
Eosinophils (1000 cells/μL)	469	0.24 (0.22)	448	0.19 (0.19)	<0.001
Basophils (1000 cells/μL)	469	0.03 (0.05)	448	0.03 (0.05)	0.746
Neutrophils (1000 cells/μL)	469	3.37 (1.51)	448	3.96 (1.71)	<0.001
WBC (1000 cells/μL)	469	6.41 (1.88)	448	6.94 (2.00)	<0.001

Data expressed as mean (standard deviation) or numbers (percentages). BMI: Body mass index; PA: Physical activity; WBC: White blood counts.

**Table 2 biology-10-00884-t002:** Association between white blood cells count and normalized handgrip strength parameters in US adolescents.

Variables	WCB (1000 Cells/μL)				
	B	SE	*p*	LLCI	ULCI
**Boys**					
Handgrip strength/Body weight					
Low	0.525	0.180	0.004	0.170	0.879
Intermediate	0.187	0.180	0.299	−0.167	0.541
High	(Ref.)				
Handgrip strength/Body fat					
Low	0.699	0.294	0.019	0.118	1.280
Intermediate	0.416	0.271	0.126	−0.118	0.950
High	(Ref.)				
Handgrip strength/Trunk fat					
Low	1.05	0.279	<0.001	0.497	1.559
Intermediate	0.444	0.265	0.095	−0.078	0.966
High	(Ref.)				
**Girls**					
Handgrip strength/Body weight					
Low	0.804	0.196	>0.001	0.419	1.189
Intermediate	0.101	0.197	0.609	−0.286	0.488
High	(Ref.)				
Handgrip strength/Body fat					
Low	1.573	0.280	<0.001	1.021	2.125
Intermediate	0.739	0.267	0.006	0.213	1.265
High	(Ref.)				
Handgrip strength/Trunk fat					
Low	1.501	0.271	<0.001	0.967	2.035
Intermediate	0.844	0.257	0.001	0.337	1.350
High	(Ref.)				

Adjusted by age, race/ethnicity, ratio of family income to poverty, physical activity, sedentary behavior and dietary intake. High normalized handgrip strength of all different groups was selected as reference category.

## Data Availability

The dataset analyzed for this study can be found in: https://www.cdc.gov/nchs/nhanes/index.htm (accessed on 8 August 2021).

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
