# Peer review of "Handgrip Strength and Its Relationship with White Blood Cell Count in U.S. Adolescents"

_biology, 2021, doi:10.3390/biology10090884_

Round 1
Reviewer 1 Report
Dear Editor and authors, I read with interest the manuscript which the main aim of this study was to clarify the association between NHS and total WBC count.
-The introduction is very well written.
-The methods: This is an important concern about the manuscript. Although, WBC count did not show a normal distribution, ANCOVA test did not reflect an adequate association between variables, in particular for NHANES data. Several manuscripts are using regression analyses adopting the p-trend following an ANCOVA for graphs. In addition, adjustment for confounding variables are imperative to reach a conclusion.
-The results: Indeed, theses data must be rewritten using the correct statistical analysis and table 1 does not make sense. The adequate is use a tertil 1-3 to show these data, divided by boys and girls.
-In addition, previous diseases and watching TV time are crucial to add as confunding variables.
The main aim of this study was to clarify the association between NHS and total WBC 98 count in a nationally representative sample of U.S. adolescents
Reviewer 2 Report
Why methods are written after results? its very weird and you need to go back and up to understand it
methods can be improved if you add headlines. Explain by groups, body composition, muscular strength, blood collection
Round 2
Reviewer 1 Report
Accept